# Learning Agile Skills via Adversarial Imitation of Rough Partial Demonstrations

**Chenhao Li**[1,2]**, Marin Vlastelica**[1]**, Sebastian Blaes**[1]**, Jonas Frey**[2,1]**,**
**Felix Grimminger**[1]**, Georg Martius**[1]
[1]Max Planck Institute for Intelligent Systems, Germany
[2] Robotic Systems Lab, ETH Zurich, Switzerland
`chenhao.li@tuebingen.mpg.de`

**Abstract:** Learning agile skills is one of the main challenges in robotics. To this end, reinforcement learning approaches have achieved impressive results. These methods require explicit task information in terms of a reward function or an expert that can be queried in simulation to provide a target control output, which limits their applicability. In this work, we propose a generative adversarial method for inferring reward functions from partial and potentially physically incompatible demonstrations for successful skill acquirement where reference or expert demonstrations are not easily accessible. Moreover, we show that by using a Wasserstein GAN formulation and transitions from demonstrations with rough and partial information as input, we are able to extract policies that are robust and capable of imitating demonstrated behaviors. Finally, the obtained skills such as a backflip are tested on an agile quadruped robot called Solo 8 and present faithful replication of hand-held human demonstrations.

**Keywords:** Adversarial, Imitation Learning, Legged Robots

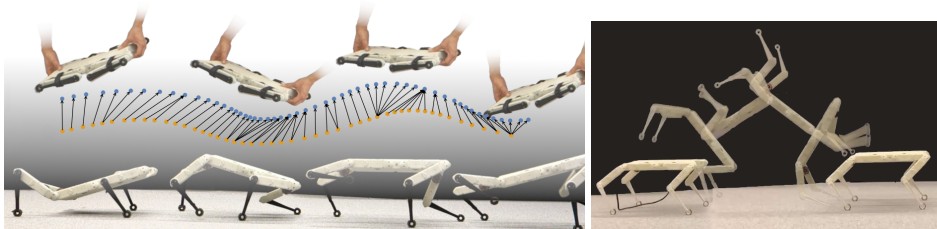

Figure 1: Our method (WASABI) achieves agile physical behaviors from rough (hand-held) and partial (robot base) motions. The illustrated performance measure is the Dynamic Time Warping distance of the base trajectories (left). A learned backflip policy is deployed on Solo 8 (right).

## 1 Introduction

Obtaining dynamic skills for autonomous machines has been a cardinal challenge in robotics. In the field of legged systems, many attempts have been made to attain diverse skills using conventional inverse kinematics techniques [1, 2]. In recent years, learning-based quadrupedal locomotion has been achieved by reinforcement learning (RL) approaches to address more complex environments and improve performance [3, 4, 5, 6]. However, the demand for acquiring more highly dynamic motions has brought new challenges to robot learning. A primary shortage of motivating desired behaviors by reward engineering is the arduous reward-shaping process involved. It can sometimes become extremely demanding in developing highly dynamic skills such as jumping and backflipping, where various terms of motivation and regularization require elaborated refinement.

Given the availability of some expert references, one possible solution is Imitation Learning (IL), which aims to mimic expert behaviors in a given task. In this framework, the agent is trained to

6th Conference on Robot Learning (CoRL 2022), Auckland, New Zealand.

perform a task from demonstrations by learning a mapping between observations and actions with either offline (e.g. behavioral cloning [7, 8]) or interactive (e.g. DAgger, SMILe [9]) methods. Generic IL methods could potentially reduce the problem of teaching a task to that of providing demonstrations, without the need for explicit programming or designing reward functions specific to the task [10]. Another related approach to replicating exerted motions of an expert is Inverse Reinforcement Learning (IRL). In IRL, the expert reward function is inferred given its policy or observed behaviors [11, 12, 13]. IRL is in general computationally expensive, and efforts are required to deal with ambiguous reward functions without making strong assumptions [14].

More recently, Generative Adversarial Imitation Learning (GAIL) [15] draws a connection between IL and generative adversarial networks (GANs) [16], which train a generative model (generator) by having it deceive a discriminative classifier (discriminator). The task of the discriminator is to distinguish between data generated by the generator and the true data distribution. In the setting of GAIL, the true data distribution is the expert state-action distribution, while the learned policy is treated as the generator. The output of the discriminator can then be used as a reward that encourages the learning agent to generate similar behaviors to the demonstration. Analogously, the technique has been used for learning adversarial motion priors (AMP) [17], where the output of the discriminator is used as an additional style reward to the actual task reward, that is available beforehand. In a sense, AMP enables solving well-defined tasks in a specific style specified by a reference motion, without requiring access to underlying expert actions.

In this work, we present a novel adversarial imitation learning method named Wasserstein Adversarial Behavior Imitation (WASABI). We show that we are able to extract sensible task rewards from rough and partial demonstrations by utilizing adversarial training for obtaining agile skills in a sim-to-real setting. In contrast to Peng et al. [17], our approach does not require any prior information about the task at hand in form of a specific reward function, but only reasonable task-agnostic regularization terms in addition to the adversarial reward that make the robot motion more stable. Most importantly, we achieve this without having access to samples from an expert policy, but rather hand-held human demonstrations that are physically incompatible with the robot itself. To the best of our knowledge, this is the first time that highly dynamic skills are obtained from limited reference information. In summary, our contributions include: **(i)** An adversarial approach for learning from partial, physically incompatible demonstrations. **(ii)** Analysis of the Least-Squares vs. Wasserstein GAN loss for reward inference. **(iii)** Experimental validation in simulation and on a quadruped robot. Supplementary videos for this work are available at `https://sites.google.com/view/corl2022-wasabi/home`.

## 2   Related Work

Advances in robotics have spawned many potential applications that require intelligent systems to be able to not only make decisions but also to perform physical movements expectedly. However, in many cases, the desired behavior may not be discovered by a learning agent due to sub-optimal parameter settings or algorithmic limitations [18, 19]. While learning a task might be stated as an optimization problem, it has become widely accepted that having prior knowledge provided by an expert is more effective and efficient than attempting to solve the problem from scratch [20, 21].

The idea of IL has been formed decades ago, raising solutions in conceptual and computational models to replicate motions from demonstrations [22, 23, 24]. It has been commonly acknowledged that IL entails three major approaches: model-based IL, learning a control policy directly, and learning from demonstrated trajectories. In the first approach, algorithms are applied to learn the parameters of the dynamics model to ensure that all executed motions closely follow the demonstration [25, 26, 27]. In the second approach, also known as behavioral cloning, the agent tries to reproduce the observed state-action pairs of the expert policy [7, 8]. Behavioral cloning often faces the problems of error compounding and poor generalization, which can lead to unstable policy output, particularly in out-of-distribution regions [28]. Alternatively, reference motions can be learned using an imitation goal, which is often implemented as a tracking objective that aims to reduce the pose error between the simulated character and target poses from a reference motion [29, 30, 31, 32]. A common strategy to estimate the pose error is to use a phase variable as an additional input to the controller to synchronize the agent with a specific reference motion [33, 32, 34]. This method typically works well for replicating single motion clips, but it may fail to scale to datasets with multiple reference motions which may not be synchronized and aligned according to a single-phase variable [17].

Instead of employing a handcrafted imitation objective, adversarial IL techniques train an adversarial discriminator to distinguish between behaviors generated by an agent and demonstrations [14, 15]. While these methods have shown some promise for motion imitation tasks [35, 36], adversarial learning algorithms are notoriously unstable, and the resulting motion quality still lags well behind that of state-of-the-art tracking-based systems. Especially in the low-data regime, adversarial models can take a long time to converge [37, 17]. In some cases, adversarial IL techniques show limited robustness against different environment dynamics, as it fails to generalize to tasks where there is considerable variability in the environment from the demonstrations [38].

With the ability to encompass multiple reference motions, AMP decouples task specification from style specification by combining GAIL with extra task objectives [17, 39]. The use of AMP reduces efforts in the selection of distance error metrics, phase indicators, and appropriate motion clips. This allows the learning agent to execute tasks that may not be portrayed in the original demonstrations. To enable active style control, Multi-AMP allows for the switching of multiple different style rewards by training multiple discriminators encoding different reference motions in parallel [40].

## 3 Approach

In this section, we describe our method, WASABI, which involves generative adversarial learning of an imitation reward from rough and partial demonstrations using a GAN framework.

### 3.1 Learning Task Reward from Limited Demonstration Information

We consider *partial* demonstrations that are given in terms of limited state observations, for instance only local velocities of the robot's base. The demonstrations are formulated as sequences of $o_t \in \mathcal{O}$, where the full state space $\mathcal{S}$ of the underlying Markov Decision Process can be mapped to the observation space $\mathcal{O}$ with a function $\Phi : \mathcal{S} \to \mathcal{O}$. We utilize generative adversarial learning for inferring the task reward function from such demonstrated transitions $(o, o')$ in a reference motion. As such, the discriminator in this setup is to distinguish samples of the policy transition distribution $d^\pi$ from the reference motion distribution $d^\mathcal{M}$. The policy $\pi$ takes on the role of the generator.

The original GAN min-max loss (CEGAN) formulation has shown to suffer from vanishing gradients due to saturation regions of the cross-entropy loss function which slows down training [41]. If the discriminator performs excessively well and thus becomes saturated, the policy will not be able to learn any information, since it receives a constant penalty for being far away from the demonstrations. For this reason, Peng et al. [17] propose to use the least-squares GAN (LSGAN) loss [42] in AMP as a substitute for reward function learning. The LSGAN loss is formulated as

$$\arg\min_{D} \mathbb{E}_{d^\mathcal{M}} \left[ \left( D(o, o') - 1 \right)^2 \right] + \mathbb{E}_{d^\pi} \left[ \left( D(\Phi(s), \Phi(s')) + 1 \right)^2 \right]. \tag{1}$$

The discriminator is defined as a mapping $D : \mathcal{O} \times \mathcal{O} \mapsto \mathbb{R}$ and can be used, together with $\Phi$, as a drop-in replacement for the unknown reward function $r(s, s')$. Intuitively, the LSGAN loss forces the discriminator to output $+1$ for samples from the reference motion and $-1$ for those from the policy. It not only prevents vanishing gradients but also provides a well-scaled output that eases downstream policy learning. However, when faced with demonstrations that initially seem beyond what the agent can achieve, the discriminator is prone to be driven to optimality, prohibiting a more fine-grained evaluation of the policy transitions with respect to their closeness to the reference motion. Moreover, the LSGAN discriminator output does not directly lead to a practical reward function by itself, since an increase in its value does not always represent close replications of demonstrated transitions. This is a consequence of the least-squares loss symmetricity around $-1$ and $+1$, therefore a suitable mapping is typically needed to transform the output into a well-behaved reward function. For this reason, we propose to use the Wasserstein loss

$$\arg\min_{D} -\mathbb{E}_{d^\mathcal{M}} \left[ D(o, o') \right] + \mathbb{E}_{d^\pi} \left[ D(\Phi(s), \Phi(s')) \right], \tag{2}$$

especially for highly dynamic motions where the discriminator is more likely to optimally distinguish between the reference and the generated motions. Under conditions of Lipschitz continuity, the Wasserstein loss is an efficient approximation to the earth mover's distance which effectively measures the distance between two probability distributions [43]. In the original Wasserstein GAN (WGAN), Arjovsky et al. [44] enforce Lipschitz continuity by projected gradient descent, i.e. clipping the

network weights. Similarly, we apply $L_2$ regularization on the discriminator for the sake of simplicity. In addition, discriminator weight regularization also controls the scale of its output, which results in stable imitation rewards.

## 3.2 Preventing Mode Collapse in Adversarial Reward Learning

Mode collapse is a common problem in GAN training, which manifests itself by the generator being able to produce only a small set of outputs. In our framework, mode collapse is reflected by the policy trying to replicate only a subset of the reference motion which gives a high reward.

The Wasserstein loss can alleviate mode collapse by allowing training of the discriminator to optimality while avoiding vanishing gradients [44]. In fact, if the discriminator does not get stuck in the local minimum, it learns to reject partial behaviors on which the policy stabilizes. As a result, the policy will have to attempt something different, if possible. In addition to the implementation of the Wasserstein loss, we extend the capability of the discriminator by allowing more than one state transition as input, i.e. we extend the input to $H$ consecutive observations. Note that this is typically not applicable to CEGAN or LSGAN, as a longer horizon makes the discriminator even stronger. By taking more sequential states into account, the policy reduces its chance to resort to the same safe transition patterns that are present in the reference motion.

We denote trajectory segments of length $H$ preceding time $t$ by $o_t^H = (o_{t-H+1}, \ldots, o_t)$ for the reference observations and $s_t^H = (s_{t-H+1}, \ldots, s_t)$ for the states induced by the policy. For clarity, we omit the time index in the following. To simplify notation, we write $\Phi(s^H)$ to express that each state in $s^H$ is mapped to $\mathcal{O}$. In our experiments, we select linear and angular velocities $v, \omega$ of the robot base in the robot frame, measurement of the gravity vector in the robot frame $g$, and the base height $z$ as the observation space $\mathcal{O}$. More information on the state space and demonstration space is detailed in Suppl. B. Note that in this example, no joint information is required by the discriminator. This facilitates the process to obtain the expert motion, as one can simply move the robot base by hand along the desired trajectory without any joint actuation.

Using $H$-step inputs and a gradient penalty, Eq. 2 turns into

$$\arg\min_D w^D \left( -\mathbb{E}_{d^{\mathcal{M}}} \left[ D\left( o^H \right) \right] + \mathbb{E}_{d^{\pi}} \left[ D\left( \Phi(s^H) \right) \right] \right) + w^{GP} \mathbb{E}_{d^{\mathcal{M}}} \left[ \left\| \nabla_\Omega D\left( \Omega \right) |_{\Omega=o^H} \right\|_2^2 \right], \quad (3)$$

where the last term denotes the penalty for nonzero gradients on samples from the dataset [17]. $w^D$ and $w^{GP}$ denote the weights on the Wasserstein loss and the gradient penalty, respectively. In our experiments, they are set to $w^D = 0.5$ and $w^{GP} = 5.0$ for all tasks.

## 3.3 Reward Formulation

Despite discriminator regularization, due to the unbounded discriminator output, the scale of the reward can be arbitrary which makes it difficult to introduce additional regularization terms for stabilizing the robot motion. Therefore, we normalize the reward to have zero mean and unit variance in the policy training loop by maintaining its running mean $\widehat{\mu}$ and variance $\widehat{\sigma}^2$. With this formulation, the imitation reward is then given by

$$r^I = \frac{D\left( \Phi(s^H) \right) - \widehat{\mu}}{\widehat{\sigma}}, \quad (4)$$

where $D\left( \Phi(s^H) \right)$ denotes the output of the discriminator.

To increase policy learning efficiency, a common practice is to define a termination condition for rollouts. In our work, an instantaneous environment reset is triggered when a robot base collision against the ground is detected. Since the imitation reward has zero mean and difficult behaviors are likely to result in negative rewards initially, the policy may attempt to end the episode early. To circumvent this, a termination penalty is imposed at the last transition before a collision happens. As the normalized reward follows a distribution with zero mean and unit variance, $-5\sigma$ is a lower bound on the reward with a probability greater than $99.99\%$. We use this to derive a reasonable termination penalty, based on the geometric series, by a high-probability lower bound on the return

$$r^T = [\![ s \in \mathcal{T} ]\!] \frac{-5\sigma}{1 - \gamma}, \quad (5)$$

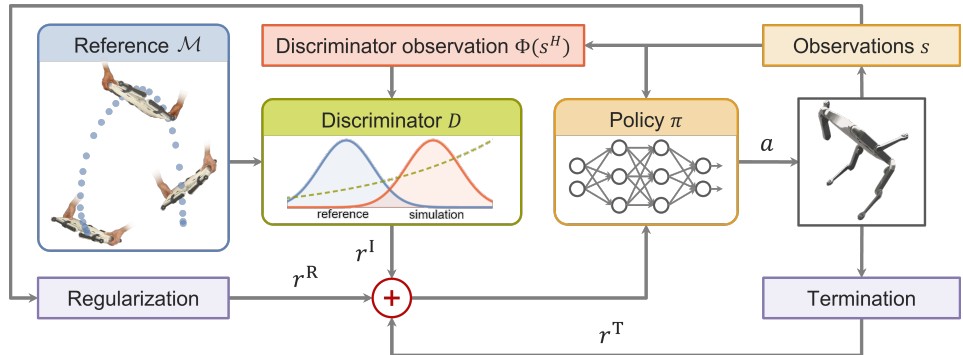

Figure 2: System overview. Given a reference dataset defining the desired base motion, the system trains a discriminator that learns an imitation reward for the policy training. This imitation reward is then combined with a regularization reward and termination penalty to train a policy that enables the robot to replicate the demonstrated motion while maintaining feasible and stable joint actuation.

where $\gamma$ is the discount factor, $\mathcal{T}$ is the set of early termination states, and $[\![\cdot]\!]$ is the Iverson bracket (1 if true, 0 otherwise). Putting everything together, the total reward that the policy receives encompasses three parts, the imitation reward $r^{\mathrm{I}}$ defined by the normalized discriminator output, the termination reward $r^{\mathrm{T}}$, and the regularization reward $r^{\mathrm{R}}$ to guarantee stable policy outputs (detailed in Suppl. C)

$$r = w^{\mathrm{I}}(r^{\mathrm{I}} + r^{\mathrm{T}}) + r^{\mathrm{R}}, \tag{6}$$

where $w^{\mathrm{I}}$ is a motion-specific scaling factor controlling the relative importance of the imitation reward (and the termination penalty) with respect to the regularization terms.

Note that our reward formulation enables the robot to learn highly dynamic skills without any explicitly defined desired-motion-incentivizing reward, as is used in AMP, where an a priori designed reward still has to motivate the policy to execute a specific movement [17]. It is also noteworthy that the LSGAN formulation in our setting can be viewed as an implementation of AMP modified for task reward learning with substantial adaptations as detailed in Suppl. D. Figure 2 provides a schematic overview of our method, and an algorithm overview is detailed in Algorithm 1.

---

**Algorithm 1** WASABI

1: **Input**: dataset of reference motions $\mathcal{M}$, feature map $\Phi$
2: initialize discriminator $D$, policy $\pi$, value function $V$, state transition buffer $s^H$, replay buffer $B$
3: **for** learning iterations $= 1, 2, \ldots$ **do**
4:     collect $N + H$ transitions $(s_t, a_t, r_t^{\mathrm{R}}, s_{t+1})_{t-H}^{t+N}$ with policy $\pi$
5:     compute $r_\tau^{\mathrm{I}}$ using discriminator outputs $D\left(\Phi(s_i^H)\right)$ for $i = t, \ldots, t + N$
6:     calculate transition rewards $r_t = w^{\mathrm{I}}\left(r_t^{\mathrm{I}} + r_t^{\mathrm{T}}\right) + r_t^{\mathrm{R}}$ according to Equations 4, 5, and 6
7:     fill replay buffer $B$ with $\left(s_t, a_t, r_t, s_{t+1}, \Phi(s_t^H)\right)_t^{t+N}$
8:     **for** policy learning epoch $= 1, 2, \ldots, n_\pi$ **do**
9:         sample transition mini-batches $b^\pi \sim B$
10:         update $V$ and $\pi$ by PPO objective or another RL algorithm
11:     **end for**
12:     **for** discriminator learning epoch $= 1, 2, \ldots, n_D$ **do**
13:         sample transition mini-batches $b^\pi \sim B$ and $b^{\mathcal{M}} \sim \mathcal{M}$
14:         update discriminator $D$ using $b^\pi$ and $b^{\mathcal{M}}$ according to the loss associated with Eq. 3
15:     **end for**
16: **end for**

---

## 4 Experiments

We evaluate WASABI on the Solo 8 robot, an open-source research quadruped robot that performs a wide range of physical actions [45], in simulation and on the real system (Fig. 3). For evaluation, we introduce 4 different robotics tasks. In SOLOLEAP, the robot is asked to move forward with a jumping motion. SOLOWAVE requires the robot to produce a wave-like locomotion behavior. For

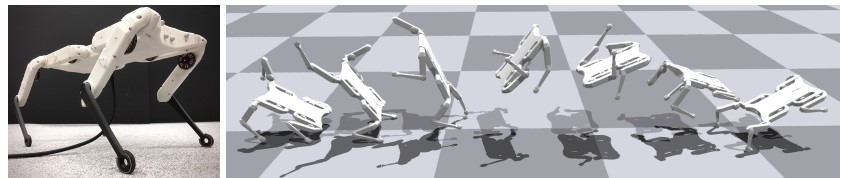

Figure 3: Solo 8 (left). Backflip motion in Isaac Gym (right).

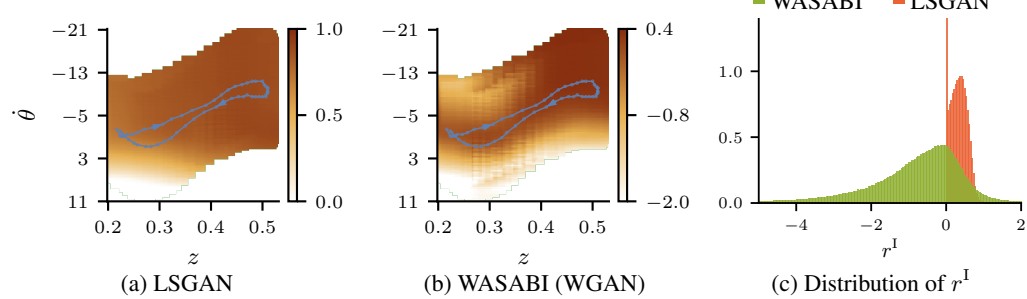

(a) LSGAN      (b) WASABI (WGAN)      (c) Distribution of $r^{\mathrm{I}}$

Figure 4: Adversarial imitation rewards for SOLOBACKFLIP. Imitation reward heatmap for LSGAN (a) and WASABI (b) around reference trajectories (blue) generated in varying pitch rate $\dot{\theta}$ and base height $z$. (c) Distribution of imitation rewards for LSGAN and WASABI during training. WASABI provides a more fine-grained reward function.

SOLOSTANDUP we require the robot to stand up on its hind legs. In SOLOBACKFLIP the robot is asked to generate motions of a full backflip. We provide *rough* demonstrations of these motions by manually carrying the robot through the motion and recording only the base information. The demonstrations are then used to infer an adversarial imitation reward for training a control policy that outputs target joint positions, as outlined in Sec. 3.1. An overview of the desired movements is provided in Suppl. G, we also provide further ablation studies in Suppl. I.

In all of our experiments, we use Proximal Policy Optimization (PPO) [46] in Isaac Gym [47] and make use of domain randomization [48] for sim-to-real transfer. Further details on the training procedure can be found in Suppl. A.

## 4.1 Induced Imitation Reward Distributions

The LSGAN loss is proposed to alleviate the saturation problem that is encountered for the CEGAN loss. Yet, as outlined in Sec. 3.1, it does not directly yield a practical reward function. Peng et al. [17] remedy this by using $r^{\mathrm{I}} = \max\left[0,\ 1 - 0.25(D\left(\Phi(s), \Phi(s')\right) - 1)^2\right]$ to map the discriminator output to the imitation reward and bound it between 0 and 1. However, with the effective clipping at 0, information about the distance from the policy to the demonstration transitions is lost with discriminator prediction smaller than $-1$ (Fig. 4c). In addition, we show in Fig. 4a that the imitation reward learned using LSGAN yields a less informative signal for policy training, which is rather uniformly distributed across pitch rate $\dot{\theta}$ and base height $z$ dimensions. In comparison, WASABI can use the discriminator output directly, learning a more characteristic reward function across the state space where reference trajectories are clearly outlined to yield high rewards in contrast to the off-trajectory states (Fig. 4b).

## 4.2 Learning to Mimic Rough Demonstrations

Since we record the base motion of the robot carried by a human demonstrator, we do not have access to a reward function evaluating learned behaviors or measuring the closeness between the demonstrated and the policy trajectories. In addition, these trajectories are largely misaligned. For this reason, we make use of Dynamic Time Warping (DTW) [49] with the $L_2$ norm metric for comparing policy trajectories and reference demonstrations. DTW allows us to match and compute the distance between the trajectories in a time-consistent manner (Fig. 1). Concretely, we use $\mathbb{E}\left[d^{\mathrm{DTW}}(\Phi(\tau_\pi), \tau_{\mathcal{M}})\right]$ as the evaluation metric, where $\tau_\pi \sim d^\pi$ is a state trajectory from a policy

| Method | SOLOLEAP | SOLOWAVE | SOLOSTANDUP | SOLOBACKFLIP |
|---|---|---|---|---|
| WASABI | **131.70 ± 16.44** | **247.29 ± 11.59** | **351.13 ± 88.60** | **477.43 ± 56.77** |
| LSGAN | **155.31 ± 18.10** | **230.91 ± 5.95** | 678.21 ± 6.71 | 813.76 ± 19.75 |
| Stand Still | 216.41 | 460.15 | 494.40 | 877.74 |

Table 1: Comparison of performances for LSGAN and WASABI trained with hand-held demonstrations in terms of **DTW distance** $d^{\mathrm{DTW}}$ (lower is better), successful runs are in **bold** font. As a reference, we provide also $d^{\mathrm{DTW}}$ of a constantly standing trajectory.

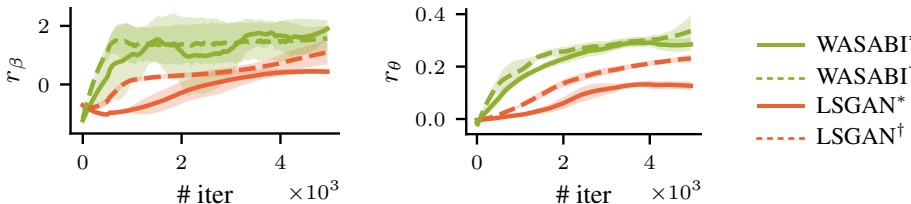

Figure 5: Performance of WASABI and LSGAN in terms of the handcrafted task reward for SOLO-STANDUP (left) and SOLOBACKFLIP (right). Dashed lines indicate partial information (†).

rollout and $\tau_{\mathcal{M}} \sim d^{\mathcal{M}}$ denotes a reference motion from the dataset. We provide further details about this metric in Suppl. H. In Table 1 we compare performances in simulation for the different reference motions.

In order to confirm that WASABI is indeed able to extract a sensible reward function that motivates the desired motion, we compare the performance of LSGAN and WASABI in SOLOSTANDUP and SOLOBACKFLIP using an expert baseline that is trained on a handcrafted task reward for generating demonstrations in simulation. Details on the handcrafted task reward formulation are given in Suppl. E. The learned policies are evaluated with the same task rewards that are used to obtain the expert policies. A comparison of training performance curves in terms of the corresponding handcrafted task rewards is detailed in Fig. 5. In Table 2 we show the performance evaluation of the best runs. Observe that the policies trained by WASABI perform comparably to the expert policies trained with the handcrafted rewards. Interestingly, learning from partial state information may sometimes facilitate policy learning, since a decrease in discriminator observation dimensions could potentially alleviate the problem of discriminator becoming too strong as indicated in Fig. 5.

### 4.3 Evaluation on Real Robot

To evaluate our method on real system, we trained policies for sim-to-real transfer with WASABI for the SOLOLEAP, SOLOWAVE and SOLOBACKFLIP. The Solo 8 robot is powered by an external battery and driven by a controller on an external operating machine. It receives root state estimation using 10 markers attached around the base which are tracked using a Vicon motion capture system operating at 100 Hz. During deployment, we recorded the robot base information for evaluation by $d^{\mathrm{DTW}}$. As detailed in Suppl. F, the policy observation space, reward, and training hyperparameters are adapted to facilitate sim-to-real transfer for these tasks specifically. The resulting performance on the real system, as shown in Table 3, resembles the performance obtained in simulation.

| Method | SOLOSTANDUP† | SOLOSTANDUP* | SOLOBACKFLIP† | SOLOBACKFLIP* |
|---|---|---|---|---|
| WASABI | **1.54 ± 0.51** | **1.68 ± 0.51** | **0.36 ± 0.05** | **0.28 ± 0.02** |
| LSGAN | 1.07 ± 0.5 | 0.44 ± 0.14 | 0.12 ± 0.01 | 0.06 ± 0.01 |
| Handcrafted | **2.24 ± 0.05** | | **0.77 ± 0.04** | |

Table 2: Performance comparison in terms of handcrafted **task reward** (higher is better). We denote with ∗ where the full robot configuration is given to the discriminator and † where only base information is given. Successful runs are in **bold** font. Std-dev. is over 5 independent random seeds.

|  | SOLOLEAP | SOLOWAVE | SOLOBACKFLIP |
|---|---|---|---|
| WASABI (Real) | $153.64 \pm 7.08$ | $215.38 \pm 21.82$ | $504.26 \pm 18.90$ |
| WASABI (Sim) | $131.70 \pm 16.44$ | $247.29 \pm 11.59$ | $477.43 \pm 56.77$ |

Table 3: Sim-to-real performance on the Solo 8 in terms of DTW distance (lower is better). Values are computed from the recorded data of the learned policies with respect to the reference trajectories.

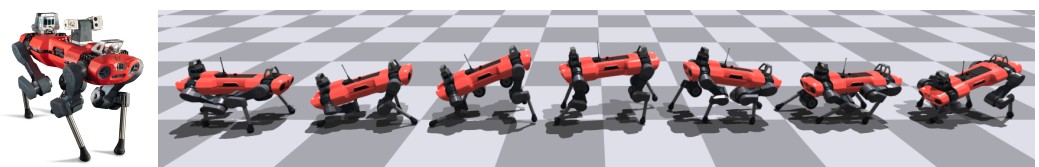

Figure 6: ANYmal C (left). Wave motion in Isaac Gym (right).

## 4.4 Cross-platform Imitation

As the reference motion in WASABI contains only base information, it does not restrict itself to be obtained only from any specific robotic platform. This provides the possibility of cross-platform imitation. Using the reference trajectories recorded from Solo 8, with a manual offset of $0.25$ m on the base height dimension addressing different sizes of the robots, we apply WASABI to ANYmal [50], a four-legged dog-like robot for research and industrial maintenance (Fig. 6). To confirm that WASABI applies to cross-platform imitation, we define ANYMALWAVE and ANYMALBACKFLIP tasks for the corresponding wave and backflip motions learned by ANYmal, yet from the reference data recorded from Solo 8. The performance in terms of the DTW distance is detailed in Table 4.

| Method | SOLOWAVE | ANYMALWAVE | SOLOBACKFLIP | ANYMALBACKFLIP |
|---|---|---|---|---|
| WASABI | $\mathbf{247.29 \pm 11.59}$ | $\mathbf{193.08 \pm 14.52}$ | $\mathbf{477.43 \pm 56.77}$ | $\mathbf{572.60 \pm 12.18}$ |
| Stand Still | 460.15 | | 877.74 | |

Table 4: Performance of cross-platform imitation of ANYmal using WASABI trained with hand-held demonstrations from Solo 8 in terms of **DTW distance** $d^{\mathrm{DTW}}$, successful runs are in **bold** font.

## 5 Conclusion

In this work, we propose an adversarial imitation method named WASABI for inferring reward functions that is capable of learning agile skills from partial and physically incompatible demonstrations without any a priori known reward terms. Our results indicate that WASABI allows extracting robust policies that are able to transfer to the real system and enables cross-platform imitation. Furthermore, our experiments confirm that imitation learning using the LSGAN fits style transfer settings where desired motions are more achievable. For highly agile or incompatible motions which initially seem beyond the robot's capability, WASABI outperforms LSGAN by successful and faithful replication of roughly demonstrated behaviors. Further extensions and applications are presented in Suppl. J.

## 6 Limitations

While saving the effort of developing a specific task reward that motivates desired motions, providing a good evaluation metric in terms of a distance to the reference motion is not straightforward for generic rough demonstrations. Although DTW is a feasible option, it still requires a reasonable distance metric and careful choice of the warping procedure, which might be task-dependent. Moreover, since our method works with rough demonstrations, even a good distance metric to the reference may not help inform about closeness to feasible, desirable motions from the robot's perspective. Finally, we do not intensively study to what extent our method is robust against the degree of incompatibility of the demonstrations.

**Acknowledgments**

Georg Martius is a member of the Machine Learning Cluster of Excellence, EXC number 2064/1 – Project number 390727645. We acknowledge the support from the German Federal Ministry of Education and Research (BMBF) through the Tübingen AI Center (FKZ: 01IS18039B). The authors thank the International Max Planck Research School for Intelligent Systems (IMPRS-IS) for supporting Marin Vlastelica and Sebastian Blaes, and Max Planck ETH Center for Learning Systems for supporting Jonas Frey.

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
