# OpenReview forum: "Learning Agile Skills via Adversarial Imitation of Rough Partial Demonstrations"
_robot-learning.org/CoRL/2022/Conference — CoRL 2022 Oral_

### Official Review · Reviewer_V4vA · 2022-07-15

**Originality:** Good
**Technical Quality:** Very Good
**Clarity Of Presentation:** Excellent
**Impact:** 4

**Recommendation:**

Strong Accept: I recommend accepting the paper and will argue for my recommendation even if other reviewers hold a different opinion.

**Summary:**

Authors present a novel generative adversarial method named WASABI for inferring reward functions from partial and potentially physically incompatible demonstrations. WASABI simultaneously trains Wasserstein GAN to distinguish between real and generated observations, and policy which uses output of the discriminator as a reward. This approach allows to use partial demonstrations which don't contain full information about the state. Authors experimentally compare performance of WGAN and LSGAN for reward inference and experimentally validate the proposed method in simulation and on a quadruped robot training complex skills such as a backflip.


**Issues:**

It seems important to mention how much expert data was used to train discriminator for each of the tasks

**Quality Of The Limitations Section:**

Limitations are addressed clearly

**Reviewer Expertise:**

4: The reviewer is confident but not absolutely certain that the evaluation is correct

**Robotics Focus:**

Sufficient demonstration on hardware

**Strengths And Weaknesses:**

The paper proposes an extension of Generative adversarial imitation learning method (GAIL) method. Wasserstein GAN formulation allows to use the discriminator output as a reward function directly only with simple normalization. This method allows to train a policy for complex behaviour from partially observed expert demonstrations. The paper is  well written and enjoyable to read.

**Summary Of Recommendation:**

It would be interesting to see ablation on how much expert data needed to train a discriminator

---

> ### Author Response · Authors · 2022-08-22
> **Response to Reviewer V4vA**
>
> Thank you for your time reviewing our work and your valuable feedback. We have improved our paper based on your concerns, as addressed in the following. Please also check the **General Response**, where we updated the paper with the improvements and presented new materials.
>
>
> **Issues**
>
> > It seems important to mention how much expert data was used to train discriminator for each of the tasks
>
> We appreciate the reviewer’s highlight of this information. For the evaluation of LSGAN and WASABI using the handcrafted reward, the policies learn from 1000 sampled trajectories from the handcrafted experts in simulation (Suppl. E). For learning from hand-held demonstrations, we use 20 reference trajectories for each of the tasks as noted in Suppl. G.1.
>
> In addition, we are happy to add the information on the number of frames for each recorded trajectory accordingly (Table S8). Furthermore, we performed an ablation study on the number of trajectories (1, 5, 10, 20) present in the reference dataset to address the reviewer’s concern. The result revealed a better performance with more expert data and the robot shows strong overfitting to potentially very poor reference when the dataset is small.

---

> > ### Comment · Reviewer_V4vA · 2022-08-27
> > **Thank you**
> >
> > Thanks for adding the ablation study. I think it increases value of the paper

---

### Official Review · Reviewer_V8R4 · 2022-07-23

**Originality:** Good
**Technical Quality:** Very Good
**Clarity Of Presentation:** Good
**Impact:** 4

**Recommendation:**

Weak Accept: I recommend accepting the paper, but will not argue for my recommendation if the majority of other reviewers have a different opinion.

**Summary:**

In this paper, the authors proposed a GAN-based framework to train agile locomotion policies using partial demonstrations. Similar to AMP (Adversarial Motion Prior) work, the authors use Wasserstein-GAN to discriminate between the expert demonstrations and the learned policies, to generate the reward signal for reinforcement learning. Different from previous works, the demonstrations in this work are not from experts (dogs for examples). Instead, only rough base movement demonstrations are provided for the learner policy to figure out dynamically feasible motions that can realize the base trajectories. The authors has demonstrated successful deployment of the learned policies on the hardware with zero-shot sim-to-real transfer.


**Issues:**

Here are some additional comments, in no particular order:

1, There are also other AMP extension works on hardware to refer to, such as this one: “Adversarial Motion Priors Make Good Substitutes for Complex Reward Functions”

2, Running mean and variance for the discriminator: Are they learnable parameters of a gaussian output?

3, Tabel 2 bottom line, why only two columns?

4, Action space for the policy is better to be included in  the main text. Also, please list the joint PD gains in the supplementary text.

5, Have the authors tried to use different observation spaces for the control policy than the discriminator? For example, base linear velocity is often not needed for the control policy to function properly. And in this way the policy may transfer to the real hardware without motion capture.

6, Why is there no SOLOSTANDUP task on the real robot? Does it transfer or not?

7, Figure 4A and 4B are not quite clear, maybe due to the color coding.


**Quality Of The Limitations Section:**

Limitations are addressed clearly

**Reviewer Expertise:**

5: The reviewer is absolutely certain that the evaluation is correct and very familiar with the relevant literature

**Robotics Focus:**

Sufficient demonstration on hardware

**Strengths And Weaknesses:**

The strengths of the paper:

1, The authors show that it is possible to only provide base trajectories as demonstrations to train imitation policies for quadruped robots. This opens many new potentials for quadruped robots, since expert demonstrations are often hard to obtain.

2, There is a good ablation study and analysis on the different GAN losses (least square vs Wasserstein)

3, The learned policy can transfer to the real hardware on various agile tasks including back flipping.

The weaknesses of the paper:

1, the delta between AMP and this work is not quite large, especially since AMP has also been tested on hardware recently. The main differences seem to be (1) whether or not to include the joint angles as part of the discriminator observations for reward generation and (2) use of WGAN vs LSGAN.

2, Not enough baselines to compare with. I think it will be great to compare AMP with the proposed method on the selected tasks.


**Summary Of Recommendation:**

See the strengths and weakness section.

---

> ### Author Response · Authors · 2022-08-22
> **Response to Reviewer V8R4 - Part 1**
>
> Thank you for your time reviewing our work and your valuable feedback. We have improved our paper based on your concerns, as addressed in the following. Please also check the **General Response**, where we updated the paper with the improvements and presented new materials.
>
>
> **Weaknesses**
>
> > 1, the delta between AMP and this work is not quite large, especially since AMP has also been tested on hardware recently. The main differences seem to be (1) whether or not to include the joint angles as part of the discriminator observations for reward generation and (2) use of WGAN vs LSGAN.
>
> > 2, Not enough baselines to compare with. I think it will be great to compare AMP with the proposed method on the selected tasks.
>
> We appreciate the reviewer for pointing out the relation of our work with the AMP paper. And we are happy to provide more information on the connections and relations between our work and the AMP work. To this end, we added Suppl. D with the following explanation and point to it at the end of the introduction of our method (line 184-186).
>
> Although both WASABI and AMP utilize generative adversarial learning methods to learn motions from reference demonstrations, from a high-level view, WASABI aims to solve fundamentally different tasks as opposed to AMP. In short, we term the **AMP method adapted for tasking learning** as *LSGAN* in our paper.
>
> Note that AMP itself does not target task reward learning. In AMP, the GAN model is employed to shape the *styles* of a learning agent while performing some other tasks, which are relatively straightforward and motivated by additional task reward functions (e.g. moving forward, reaching a target). This can be viewed as learning the regularization of motions. Indeed, we can adapt AMP to enable it to directly learn complex *tasks* by providing the desired motions as the reference and removing the original task reward. In addition, to alleviate the mode collapse issue, the capability of the discriminator is extended to encompass longer observation horizons. With these adaptations, AMP is referred to as LSGAN in our work.
>
> Typically, AMP learns demonstrated motions with sufficient exploration enabled by reference state initialization (RSI), where the agent is initialized randomly along the reference trajectories. However, in the setting of legged skill learning from only base information, RSI is not applicable due to the missing joint reference. This makes the robot difficult to directly imitate highly dynamic motions. And as shown in Table 1, the robot fails to adopt complex skills (e.g. SOLOSTANDUP, SOLOBACKFLIP) with the LSGAN implementation.
>
> In contrast, WASABI is proposed to learn *task* rewards directly for agile motions from limited data. This becomes especially meaningful for tasks where the reward function is challenging to design and a decent expert is not immediately accessible. WASABI provides solutions to cases where we want to quickly develop a complex skill for robot learning, allowing us to hand-hold a robot (or an object) without actuating it and to learn directly from the demonstrated trajectories.
>
> **Issue 1**
>
> > 1, There are also other AMP extension works on hardware to refer to, such as this one: “Adversarial Motion Priors Make Good Substitutes for Complex Reward Functions”
>
> We thank the reviewer for pointing us in the direction of this mentioned paper. It is indeed closely related to our work, and we update our manuscript to include this citation in line 93.
>
> **Issue 2**
>
> > 2, Running mean and variance for the discriminator: Are they learnable parameters of a gaussian output?
>
>
> In our setting, they are not learnable parameters of a Gaussian output. Instead, the mean and variance are updated at every batch of the discriminator training using the [parallel algorithm](https://en.wikipedia.org/wiki/Algorithms_for_calculating_variance#Parallel_algorithm).
>
> **Issue 3**
>
> > 3, Tabel 2 bottom line, why only two columns?
>
> “Handcrafted” refers to the expert policies that are developed with the handcrafted rewards (Suppl. E) using plain reward shaping. The values in the two columns denote the evaluation of the experts using these handcrafted rewards. At this stage, it does not yet include any imitation learning and thus does not distinguish learning between full and partial reference information. We provide this information as the upper bound for the performance of the policies learned from imitation.

---

> ### Author Response · Authors · 2022-08-22
> **Response to Reviewer V8R4 - Part 2**
>
> **Issue 4**
>
> > 4, Action space for the policy is better to be included in the main text. Also, please list the joint PD gains in the supplementary text.
>
> We thank the reviewer for pointing this out. We added the policy output (desired joint positions) to line 197 of the main text and PD gains to line 43 of the supplementary text.
>
> **Issue 5**
>
> > 5, Have the authors tried to use different observation spaces for the control policy than the discriminator? For example, base linear velocity is often not needed for the control policy to function properly. And in this way the policy may transfer to the real hardware without motion capture.
>
> The observation spaces for the discriminator (Table S4) and the policy (Table S5) are indeed different. If the reviewer means the base part specifically, we tried to remove base height from the policy observation space and the learned policies still worked well for all four tasks. Including base state estimation in the observation space of the control policy is a common practice in robotic control [[1](https://www.science.org/doi/10.1126/scirobotics.aau5872), [2](https://www.science.org/doi/10.1126/scirobotics.abc5986), [3](https://www.science.org/doi/10.1126/scirobotics.abk2822)]. Although investigating observation space representation does not belong to the main scope of the paper, we appreciate the reviewer for raising the possibility of getting rid of the external motion tracking system by relying only on the measurement of only the joint information.
>
> Therefore, we performed experiments to test this idea. For locomotion tasks (SOLOLEAP, SOLOWAVE), the robot still manages to imitate the reference motion. However, for more dynamic tasks (SOLOSTANDUP, SOLOBACKFLIP), the policy fails to learn without the perception of the base state. In this case, the robot fails to distinguish states where it has to react differently from only the joint information. For instance, perceiving only the joint states does not provide enough information about how stably the robot stands in SOLOSTANDUP or whether the robot has successfully taken off in SOLOBACKFLIP. To illustrate this, we added a [policy observation ablation video](https://youtu.be/8OaDqDstqIs) for both SOLOWAVE and SOLOBACKFLIP on [our website](https://sites.google.com/view/corl2022-wasabi/home).
>
> **Issue 6**
>
> > 6, Why is there no SOLOSTANDUP task on the real robot? Does it transfer or not?
>
> We tried to deploy the learned policy from either the handcrafted expert or obtained by WASABI for SOLOSTANDUP on the real robot. It turned out that the robot failed to maintain its balance after it managed to stand up. As Solo 8 does not have Hip Abduction Adduction (HAA) joints, it could be challenging to adapt against unexpected base rolling accordingly, which happens frequently after standing up. This makes this task especially difficult. In addition, the sim-to-real gap could also be potentially encoded in the inaccurate modeling of the point contacts between the feet and the ground, which the robot requires to adapt constantly to maintain its attitude. We acknowledge the reviewer’s concern and added the explanation also to Suppl. F.4 in the paper.
>
> **Issue 7**
>
> > 7, Figure 4A and 4B are not quite clear, maybe due to the color coding.
>
> Thanks for pointing this out. We adapted the figures to make them more visible.
>
>
> [[1](https://www.science.org/doi/10.1126/scirobotics.aau5872)] Hwangbo, J., Lee, J., Dosovitskiy, A., Bellicoso, D., Tsounis, V., Koltun, V. and Hutter, M., 2019. Learning agile and dynamic motor skills for legged robots. *Science Robotics*, *4*(26), p.eaau5872.
>
> [[2](https://www.science.org/doi/10.1126/scirobotics.abc5986)] Lee, J., Hwangbo, J., Wellhausen, L., Koltun, V. and Hutter, M., 2020. Learning quadrupedal locomotion over challenging terrain. *Science Robotics*, *5*(47), p.eabc5986.
>
> [[3](https://www.science.org/doi/10.1126/scirobotics.abk2822)] Miki, T., Lee, J., Hwangbo, J., Wellhausen, L., Koltun, V. and Hutter, M., 2022. Learning robust perceptive locomotion for quadrupedal robots in the wild. *Science Robotics*, *7*(62), p.eabk2822.

---

### Official Review · Reviewer_Hona · 2022-07-28

**Originality:** Very Good
**Technical Quality:** Good
**Clarity Of Presentation:** Fair
**Impact:** 3

**Recommendation:**

Weak Accept: I recommend accepting the paper, but will not argue for my recommendation if the majority of other reviewers have a different opinion.

**Summary:**

The paper proposed an adversarial imitation learning method to learn agile motor skills of a quadrupedal robot from partial demonstrations.
The main contributions of the paper are following two points:
1. The authors introduced the Wasserstein loss to the adversarial reward learning.
2. The authors showed objective and reward functions to acquire the agile motor behaviors from the rough and partial demonstrations.


**Issues:**

* I suggest the authors to improve the paper structure so that, ideally, the paper becomes understandable without reading the original AMP paper.
* I suggest the authors to perform the ablation study for the proposed reward function Eq. (6) to show that both $r^{T}$ and $r^{R}$ are important.


**Quality Of The Limitations Section:**

Limitations are addressed clearly

**Reviewer Expertise:**

3: The reviewer is fairly confident that the evaluation is correct

**Robotics Focus:**

Sufficient demonstration on hardware

**Strengths And Weaknesses:**

Strength:
* The authors successfully demonstrated that the proposed method, Wasserstein Adversarial Behavior Imitation (WASABI), could learn the agile behaviors from the handcrafted partial state information.
* The authors successfully showed that the agile behaviors of not only the simulated but also real quadrupedal robot could be generated with WASABI by transferring the learned policies.

Weakness:
* Although the experimental results are fascinating, the paper seems difficult to read. Especially, readers may need to read the original paper of adversarial motion priors (AMP) [Peng et. al., TOG, 2021] to understand the learning process of the adversarial reward learning and policy training. To improve this, for example, I recommend that the authors make a new section to explain the original AMP method as a preliminary in the paper.
* The ablation study of the proposed reward function Eq. (6) is missing.

Minor comments:
* Since the backflip behavior of the real quadrupedal robot is impressive, it would be better if the authors show snapshots of the backflip on the paper like Figure 1 (left). Figure 1 (right) is hard to see.


**Summary Of Recommendation:**

The contribution of this paper is sufficient: strength in both originality of the approach and thoroughness of results. The experiments are performed in not only simulation but also real environments. However, the clarity of the presentation is insufficient. The paper should be self-contained.

---

> ### Author Response · Authors · 2022-08-22
> **Response to Reviewer Hona**
>
> Thank you for your time reviewing our work and your valuable feedback. We have improved our paper based on your concerns, as addressed in the following. Please also check the **General Response**, where we updated the paper with the improvements and presented new materials.
>
>
> **Weakness 1, Issue 1**
>
> > I suggest the authors to improve the paper structure so that, ideally, the paper becomes understandable without reading the original AMP paper.
>
> We value the reviewer’s feedback and we changed the paper structure to improve the readability of the paper without knowledge of the original AMP paper.
>
> For this reason, we adapted line 113 and tried to avoid referring to the whole AMP framework, but only introduced the specific GAN formulation (LSGAN) used in AMP. To understand the advantage of using the WGAN loss (Eq. 2) over the LSGAN loss (Eq. 1), readers are not expected to have read the AMP paper.
>
> We also appreciate the reviewer for pointing out the relation of our work with the AMP paper. And we are happy to provide more information on the connections and relations between our work and the AMP work. To this end, we added Suppl. D and point to it at the end of the introduction of our method (line 184-186).
>
> **Weakness 2, Issue 2**
>
> > I suggest the authors to perform the ablation study for the proposed reward function Eq. (6) to show that both rT and rR are important.
>
> The termination reward $r^\mathrm{T}$ is implemented to avoid early episode termination by the potential negative value of the imitation reward $r^\mathrm{I}$. Without $r^\mathrm{T}$, the policy learns to directly collide with the ground to reset itself (Sec. 3.3). The regularization reward $r^\mathrm{R}$ is a common practice for robotic control using reinforcement learning methods to guarantee safe and stable motions [[1](https://www.science.org/doi/10.1126/scirobotics.aau5872), [2](https://www.science.org/doi/10.1126/scirobotics.abc5986), [3](https://www.science.org/doi/10.1126/scirobotics.abk2822)]. Without $r^\mathrm{R}$, the task can still be learned. However, the generated motions can be very inconsistent and aggressive (e.g. sudden change of joint positions, high-frequency vibrations). We appreciate the reviewer for pointing this out. To illustrate the importance of the reward components, we added reward ablation videos for both the [termination](https://youtu.be/mNguf2FhGdg) and the [regularization](https://youtu.be/7dbmZInA5uU) terms on [our website](https://sites.google.com/view/corl2022-wasabi/home).
>
> **Minor Comments**
>
> > Since the backflip behavior of the real quadrupedal robot is impressive, it would be better if the authors show snapshots of the backflip on the paper like Figure 1 (left). Figure 1 (right) is hard to see.
>
> We appreciate the reviewer’s acknowledgment of the backflip motion on the real robot. Indeed we considered switching Figure 1 (left) to the backflip motion. However, in addition to space limitations to hold a vertical motion, we thought the point mapping in the figure illustrating the Dynamic Time Warping (DTW) algorithm might not be clearly visible in the backflip motion as the trajectories of taking off and landing heavily overlap. But we are happy to try if we could get more space for the camera-ready version. Regarding Figure 1 (right), we updated it for better visibility.
>
>
> [[1](https://www.science.org/doi/10.1126/scirobotics.aau5872)] Hwangbo, J., Lee, J., Dosovitskiy, A., Bellicoso, D., Tsounis, V., Koltun, V. and Hutter, M., 2019. Learning agile and dynamic motor skills for legged robots. *Science Robotics*, *4*(26), p.eaau5872.
>
> [[2](https://www.science.org/doi/10.1126/scirobotics.abc5986)] Lee, J., Hwangbo, J., Wellhausen, L., Koltun, V. and Hutter, M., 2020. Learning quadrupedal locomotion over challenging terrain. *Science Robotics*, *5*(47), p.eabc5986.
>
> [[3](https://www.science.org/doi/10.1126/scirobotics.abk2822)] Miki, T., Lee, J., Hwangbo, J., Wellhausen, L., Koltun, V. and Hutter, M., 2022. Learning robust perceptive locomotion for quadrupedal robots in the wild. *Science Robotics*, *7*(62), p.eabk2822.

---

### Official Review · Reviewer_2hjT · 2022-08-06

**Originality:** Very Good
**Technical Quality:** Very Good
**Clarity Of Presentation:** Very Good
**Impact:** 4

**Recommendation:**

Weak Accept: I recommend accepting the paper, but will not argue for my recommendation if the majority of other reviewers have a different opinion.

**Summary:**

This paper presents an imitation learning algorithm, named WASABI, that enables quadruped robots to learn from rough and partial expert demonstration using generative adversarial learning. Instead of using task-specific reward functions, as with many robotics learning approaches, the proposed approach can infer rewards from demonstration using GAN. In particular, the authors incorporate task-agnostic regularization terms and adversarial rewards to enable the robot to learn stable and agile locomotion behaviors. Moreover, this approach can learn only from partial and physically incompatible demonstrations (e.g., a human operator holds and moves the robot's trunk in a specific fashion). The authors also showed simulation and experimental results of a quadruped robot learning four challenging tasks. The papers show that WASABI outperforms the least-square GAN (LSGAN) in most of the tasks (3 out of 4) in terms of motion similarity (measured by DTW distance).

**Issues:**

Please explain the reasoning behind the choice of observation speed.

**Quality Of The Limitations Section:**

Additional details required

**Reviewer Expertise:**

4: The reviewer is confident but not absolutely certain that the evaluation is correct

**Robotics Focus:**

Sufficient demonstration on hardware

**Strengths And Weaknesses:**

**Strength:**

- Able to learn from limited demonstration without defining task-specific reward functions, which often need to be designed and tuned by hand.
- Replacing LSGAN loss with Wasserstein loss improves the efficiency in discriminating the reference and the generated motions.
- Able to prevent mode collapse using gradient penalty term in the loss function.
- Demonstrating the performance of the proposed approach in hardware experiments

**Weakness:**

- The final reward function also includes a regularization reward, which seems to be specific to the robot. How much does this term contribute to the total reward?
- The observation space does not include the position/orientation of the robot base (it only includes linear and angular velocities). Will the demonstration speed of the human operator would affect the final learned locomotion?

**Summary Of Recommendation:**

Overall, this work presented in the paper is well developed. In robot learning (especially when it involves hardware), the reward design or shaping is often heuristic and requires manual tuning for different tasks. WASABI, which uses generative adversarial learning, provides a principled means to design rewards for robot learning. The demonstration of hardware shows the promising potential of the proposed approach.

---

> ### Author Response · Authors · 2022-08-22
> **Response to Reviewer 2hjT - Part 1**
>
> Thank you for your time reviewing our work and your valuable feedback. We have improved our paper based on your concerns, as addressed in the following. Please also check the **General Response**, where we updated the paper with the improvements and presented new materials.
>
> **Weakness 1**
>
> > The final reward function also includes a regularization reward, which seems to be specific to the robot. How much does this term contribute to the total reward?
>
> The robot is able to perform the learned task already even without any regularization. However, the motions can be very inconsistent and aggressive (e.g. sudden change of joint positions, high-frequency vibrations). In this case, regularization terms are expected to generate stable and smooth robot motions, as with many reinforcement learning control methods for robotics [[1](https://www.science.org/doi/10.1126/scirobotics.aau5872), [2](https://www.science.org/doi/10.1126/scirobotics.abc5986), [3](https://www.science.org/doi/10.1126/scirobotics.abk2822)]. To illustrate this, we added a video of [regularization reward ablation](https://youtu.be/7dbmZInA5uU) on [our website](https://sites.google.com/view/corl2022-wasabi/home).
>
> In the simulation, we find a set of task-agnostic regularization terms (Suppl. C, Table S6) sufficient to generate visually decent motions. But when deployed to the real system, these terms need more careful design. For the best performance, they are recommended to be specifically adjusted for different robot platforms, different tasks, and different quality of reference motions to generate safe and stable motions on the real robot (Table S7). In terms of the actual scales of these terms with respect to the imitation reward, we are happy to provide details in the following table.
>
> | Task      | $w^\mathrm{I} r^\mathrm{I}$   | $r_{ar}$ | $r_{q_a}$ | $r_{q_T}$             | $r_{\dot{\phi}}$ | $r_{\dot{\psi}}$      | $r_{\dot{y}}$         | $r_{t_f}$ |
> |:---------:|:-------:|:--------:|:---------:|:---------------------:|:----------------:|:---------------------:|:---------------------:|:---------:|
> | SOLOLEAP | $-0.17$ | $-0.01$  | $-0.05$   | $-2.6 \times 10^{-4}$ | $-0.01$          | $-3.9 \times 10^{-3}$ | $-2.0 \times 10^{-3}$ | $0.03$    |
> | SOLOWAVE | $-0.03$ | $-0.01$  | $-0.03$   | $-2.2 \times 10^{-4}$ | $-0.02$          | $-4.8 \times 10^{-3}$ | $-2.9 \times 10^{-3}$ | $0.01 $   |
> | SOLOBACKFLIP   | $0.32$  | $-0.02$  | $-0.20$   | $-1.0 \times 10^{-4}$ | $-0.02$          | $-6.4 \times 10^{-3}$ | $-2.5 \times 10^{-3}$ | $0.00$    |
>
> **Weakness 2**
>
> > The observation space does not include the position/orientation of the robot base (it only includes linear and angular velocities). Will the demonstration speed of the human operator would affect the final learned locomotion?
>
> Indeed, the observation space for both the discriminator (Table S4) and the policy (Table S5) does not include explicit position and orientation information of the robot base. But part of this information is *implicitly* encoded in the base height and projected gravity (gravity vector projected in the robot frame) dimensions.
>
> We appreciate the reviewer for pointing out the factor of demonstration speed. If the demonstration speed is *uniformly* high in the reference dataset, then the corresponding recorded velocity terms will also be high, thus the policy will try to mimic this and also operate at a high locomotion speed. If there is some variance in the velocity terms *within* the reference dataset, the policy will have larger freedom in choosing its locomotion velocity. In this case, the regularization terms may play a role in restricting this freedom (e.g. action rate and joint acceleration rewards might prefer slow locomotion over a fast one.). In our work, we collect 20 distinct reference motions for each task with some variance with different demonstration speeds. For SOLOLEAP and SOLOWAVE, we successfully enable active velocity control by adding an additional regularization term tracking a velocity command within this dataset diversity. On [our website](https://sites.google.com/view/corl2022-wasabi/home), we added a video of [active velocity control](https://youtu.be/Tst8D3bxdYU) for SOLOWAVE, taking the advantage of the velocity diversity in our recorded dataset.

---

> ### Author Response · Authors · 2022-08-22
> **Response to Reviewer 2hjT - Part 2**
>
> **Issues**
>
> > Please explain the reasoning behind the choice of observation speed.
>
> It would be appreciated if the reviewer could explain more about what is meant by “observation speed”. If the reviewer means the demonstration speed, please refer to the explanation above. For SOLOLEAP and SOLOWAVE, the robot is able to adapt to different reference velocities. For SOLOSTANDUP and SOLOBACKFLIP, no physical incompatibility (e.g. holding the robot base statically in the air) is intentionally imposed. Above all, the speed of the human demonstrations is relatively arbitrary. And we hope this idea of no specific requirements for the reference demonstrations is considered an advantage of WASABI.
>
> [[1](https://www.science.org/doi/10.1126/scirobotics.aau5872)] Hwangbo, J., Lee, J., Dosovitskiy, A., Bellicoso, D., Tsounis, V., Koltun, V. and Hutter, M., 2019. Learning agile and dynamic motor skills for legged robots. *Science Robotics*, *4*(26), p.eaau5872.
>
> [[2](https://www.science.org/doi/10.1126/scirobotics.abc5986)] Lee, J., Hwangbo, J., Wellhausen, L., Koltun, V. and Hutter, M., 2020. Learning quadrupedal locomotion over challenging terrain. *Science Robotics*, *5*(47), p.eabc5986.
>
> [[3](https://www.science.org/doi/10.1126/scirobotics.abk2822)] Miki, T., Lee, J., Hwangbo, J., Wellhausen, L., Koltun, V. and Hutter, M., 2022. Learning robust perceptive locomotion for quadrupedal robots in the wild. *Science Robotics*, *7*(62), p.eabk2822.

---

### Author Response · Authors · 2022-08-22
**General Response**

**Comment:**

We would like to thank all reviewers for their valuable feedback. Here we summarize the major changes we made based on the feedback we received from the reviewers:
- We adjusted Sec 3.1 to improve the readability without the necessity of having read the AMP work. (Reviewer Hona)
- We updated Fig. 1 (right) to improve the visibility of the backflip motion. (Reviewer Hona)
- We added Suppl. D to provide more information on the connections and relations between our work and the AMP work. (Reviewers Hona, V8R4)
- We added new citations to related work. (Reviewer V8R4)
- We added the action space of the control policy to line 197 of the main text and PD gains to line 43 of the supplementary text. (Reviewer V8R4)
- We added Suppl. F.4 to describe and analyze the performance of the SOLOSTANDUP policy deployed on the real system. (Reviewer V8R4)
- We updated Fig. 4a and Fig. 4b for better visibility. (Reviewer V8R4)
- We updated Suppl. E, Suppl. G.1. and added Table S8 to highlight the amount of reference data used in each task. (Reviewer V4vA)
- We also corrected minor typos in the main text and the supplementary.


In addition to the changes on the paper text, we uploaded additional videos on [our website](https://sites.google.com/view/corl2022-wasabi/home) addressing the following concerns:
- We added reward ablation videos to address concerns regarding the importance of the regularization and termination rewards. (Reviewers 2hjT, Hona)
- We added a velocity control video to highlight how we realize active locomotion velocity control by taking advantage of the variance of the demonstration speed by the human operator within the reference dataset. (Reviewer 2hjT)
- We added a policy observation ablation video to show learned motions after removing all base perceptions in the control policy observation space to explore the possibility of getting rid of the reliance on the external motion capture. (Reviewer V8R4)


Finally, we presented some *new extensions* of our work in Suppl. J. including the realization of a single backflip and the possibility of cross-platform skill imitation. These extensions are built based on small modifications to the adversarial imitation learning framework proposed in our work and are meant to illustrate potential applications of WASABI in different directions. The corresponding supplementary videos are added on [our website](https://sites.google.com/view/corl2022-wasabi/home).

We uploaded the updated main paper and supplementary text. In the attached ZIP file, we provide `main_diff.pdf` and `suppl_diff.pdf` to highlight the text changes we made.


**Zip File:**

/attachment/8227204bbf37e3c453ffd0dcef1b864586c1a017.zip

---

### Comment · Area_Chair_AQ42 · 2022-08-25
**Engagement with authors regarding their responses to the reviews**

Dear reviewers, the authors have responded in detail to the reviews.  Please do further engage with the authors now, if possible, as the author/reviewer discussion/rebuttal phase ends Aug 27 at 11:59 PM Pacific.
Many thanks in advance -- your participation greatly contributes to the overall quality and value of the review process!
-- your Area Chair

---

### Meta-Review · Area_Chair_AQ42 · 2022-08-15

**Recommendation:** Accept (Oral)
**Confidence:** 5

**Metareview:**

The paper proposes an extension of Generative adversarial imitation learning method (GAIL) method,
and shows how it can be used to generate a back-flip for the Solo8 quadruped, using a hand-held human demonstration,
in addition to leap, wave, and standup motions. It builds on the general approaches of GAIL and, more recently, AMP.

The paper has recommendations of:  strong accept, weak accept x3.
The remaining weaknesses and issues are thoroughly addressed in the rebuttal.
This is a paper that will be appreciated by all.  The hardware demonstrations are impressive.
My recommendation:  Accept (Oral), consider for best paper

Strengths:
- well written
- ability to learn from limited demonstrations , i.e., body trajectory sketch, without reward functions;
- important details and related ablations : LSGAN --> Wasserstein loss, gradient penalty term in loss function
- hardware demonstrations

Weaknesses:
- delta to the AMP paper is not that large;  why not compare with AMP  (now addressed in Supp. D)
- final regularization reward may still be robot-specific (reward ablation videos now added)
- paper may be difficult to read, i.e., may need to read AMP  (sec 3.1 improved in revised version)


**Best Paper Nomination:**

Yes

---

> ### Author Response · Authors · 2022-08-24
> **Response to Area Chair AQ42**
>
> Thank you for your time reviewing our work and your valuable feedback. We have improved our paper based on the concerns, as addressed in our replies to each individual reviewer. We also summarized the main changes in the separate post of **General Response**, where we updated the paper with the improvements and presented new materials.